# SHORTGPT: LAYERS IN LARGE LANGUAGE MODELS ARE MORE REDUNDANT THAN YOU EXPECT

## ABSTRACT

As Large Language Models (LLMs) continue to advance in performance, their size has increased significantly, with current LLMs containing billions or even trillions of parameters. In this study, we identify notable redundancy across the layers of LLMs, where some layers contribute minimally to overall network functionality. To quantify this, we introduce a metric called Block Influence (BI) which use the similarity between layer's input and output to measure the importance of each layer. Based on the observation of layer redundancy, we propose a straightforward pruning method: layer removal, which eliminates redundant layers based on their BI scores. Our approach, termed ShortGPT, demonstrates superior performance over previous state-of-the-art pruning methods. Moreover, ShortGPT is orthogonal to quantization-like methods, enabling further reduction in parameters and computation. The ability to achieve better results through simple layer removal, as opposed to more complex pruning techniques, suggests a high degree of redundancy across layers, not only in transformer models but also in non-transformer models. We hope this work will contribute to future research in LLM compression.

## 1 INTRODUCTION

The field of large language models (LLMs) has witnessed rapid development recently, with LLMs achieving impressive performance across various domains. Guided by the scaling laws identified in prior work (Kaplan et al., 2020; Hoffmann et al., 2022), current LLM research tend to increase model parameters to boost performance. As a result, modern LLMs, which can comprise billions to trillions of parameters, require significant hardware resources for deployment, creating substantial barriers to their practical use.

To mitigate the hardware demands of large models, model compression techniques have become a critical area of focus (Zhu et al., 2023). These techniques are generally divided into quantization (Liu et al., 2021; Gholami et al., 2022; Dettmers et al., 2022; 2024) and pruning(LeCun et al., 1989; Han et al., 2015; Frantar & Alistarh, 2023). Quantization reduces the precision of model parameters, but its effectiveness often requires specific hardware support. In contrast, pruning method removes redundant parameters to decrease the model's size and computation, offering a more flexible and hardware-agnostic approach. Despite its advantages, many existing pruning methods are complex; for example, some require gradient information (Ma et al., 2024), which limits their practicality.

In this paper, we focus on the issue of layer redundancy in LLMs and propose a novel approach for simplifying these models. We introduce **Block Influence (BI)**, a metric that quantifies how much the hidden state changes after passing through each layer, providing a more direct measure of a layer's importance. Leveraging this insight, we propose a simple yet effective pruning method **ShortGPT**, which identifies and removes layers with lower BI scores, significantly reducing model size without sacrificing much performance.

To evaluate our approach, we conducted evaluation across comprehensive benchmarks. Our experiments revealed that our method exhibits a smaller performance decrement compared to the previous methods. For instance, removing 10 layers (25% of the total 40 layers) from the LLaMA 2-13B model resulted in only a slight drop in performance on the MMLU benchmark (Hendrycks et al., 2020), from 55.0 to 52.2. Our findings highlight substantial redundancy in current LLMs and suggest

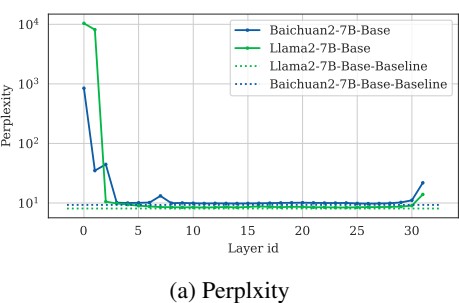 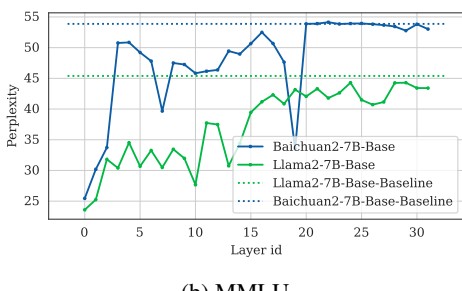

(a) Perplxity                                        (b) MMLU

Figure 1: Performance of removing certain layer from LLMs. We can see that certain layers are redundant, and their removal results in minimal performance degradation.

potential avenues for improving the efficiency of model training by reducing inherent redundancy in the future.

The main contributions of our paper are summarized as follows:

- We analyze the redundancy in large language models (LLMs) and find that they exhibit significant redundancy at the layer level. This finding inspire us to prune LLMs by simply removing redundant layers.

- We propose a metric called Block Influence (BI) as an indicator of layer importance. Based on BI, our layer removal method maintains approximately 90% performance while reducing approximately 25% of parameters, outperforming previous state-of-the-art methods.

- Furthermore, we demonstrate that our layer pruning approach is orthogonal to quantization methods, meaning it can be combined with quantization techniques to further reduce the deployment overhead of LLMs.

## 2 MOTIVATION

### 2.1 BACKGROUND

The predominant LLMs are primarily based on the Transformer architecture (Vaswani et al., 2017), with the pre-norm configuration being the most commonly adopted, as in models like LLaMA (Touvron et al., 2023). The pre-norm configuration, where layer normalization is applied before the self-attention and feed-forward layers, offers several advantages such as faster convergence, improved training stability, and better scalability for deeper networks (Xiong et al., 2020; Liu et al., 2020; Wang et al., 2024). Due to these benefits, the pre-norm approach has been adopted even in non-transformer models, such as Mamba (Gu & Dao, 2023) and RWKV (Peng et al., 2023). For the sake of simplicity in descriptions, our analysis primarily focuses on the Transformer architecture, though we extend our experiments to non-Transformer structures in Section 4.4.

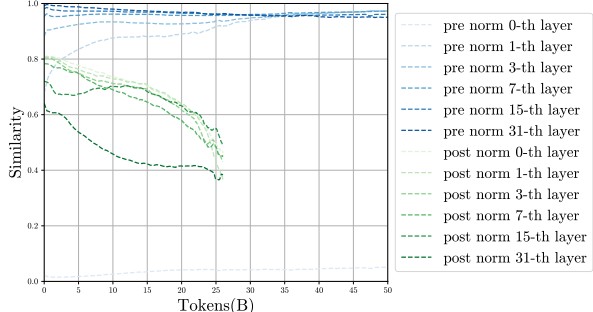

Figure 2: The cosine similarity between a layer's input and output during the training process. The horizontal axis (X-axis) represents the number of training tokens, while the vertical axis (Y-axis) depicts the degree of similarity. Notably, the model employing post-normalization exhibits divergence after approximately ∼26B tokens of training. Training setting is provided in E.

However, we observe that when pre-norm is adopted, the similarity between the input and output of transformer layers tends to be higher, as illustrated in Figure 2. This high similarity indicates that certain layers induce minimal changes to the hidden states, suggesting they contribute little to the model's overall function. A detailed mathematical explanation for this phenomenon is provided in Appendix A. Which suggests that the deep layers of the model with pre-norm might not play a critical role in the overall function, and that **the layers in large language models could be more redundant than expected**, which motivates the layer-removal based pruning method we explore in the next section.

## 2.2 LAYER REDUNDANCY

As discussed in the previous section, we speculate that the LLMs exhibit layer redundancy. To verify this, we assess the performance degradation caused by removing individual layers of two popular models, Llama2-7B-Base (Touvron et al., 2023), an English based LLMs, and Baichuan2-7B-Base (Yang et al., 2023) which is mainly focused on Chinese. Figure 1 confirms our speculation, which reveals that some layers do not play a crucial role in LLMs, causing little degradation when omitting them individually. Moreover, this redundancy is primarily manifested in the middle to later layers of the network, with the initial layers and the last layer often being more critical. Notably, we found the last layer to be

Table 1: Ablation of removing FFN and Attention of Llama2-7B-Base. We sample 100 instances from PG19 (Rae et al., 2019) to calculate PPL.

| Delete | PPL |
|---|---|
| None | 7.60 |
| The whole last layer | 13.37 |
| Attention of the last layer | 7.65 |
| FFN of the last layer | 12.35 |

particularly important, aligning with findings from LLM Pruner (Ma et al., 2024). This observation contradicts our mathematical explanation in Appendix A which suggests that deeper layers tend to be more redundant. We posit that this discrepancy arises because the final FFN effectively functions as part of the token classifier and should be considered in conjunction with the language model head. To verify our hypothesis, we conducted further investigation, detailed in Table 1. The results show that within the last layer, the FFN component is crucial, while the Attention module is less significant. This finding supports our interpretation of the final layer's importance.

## 3 METHODOLOGY

In this section, we present the methodological framework of our layer removal approach for LLMs, elucidating the underlying principles and techniques employed. We begin by introducing Block Influence (BI), a novel metric designed to assess the hidden states transformation of each layer. Leveraging BI, we then detail our layer removal method.

## 3.1 LAYER IMPORTANCE

As outlined in the preceding section, the layers of LLMs exhibit redundancy, with varying degrees of redundancy across different layers. To capture this, we introduce a new metric, Block Influence (BI), to measure the degree of transformation performed by each layer. The BI score of $i^{th}$ layer can be calculated as follows:

$$\text{BI}_i = 1 - \mathbb{E}_{X,t} \frac{X_{i,t}^T X_{i+1,t}}{||X_{i,t}||_2 ||X_{i+1,t}||_2}, \tag{1}$$

where $X_{i,t}$ means the $t^{th}$ row of hidden states of $i^{th}$ layer. Lower BI score imply that $X_i$ and $X_{i+1}$ exhibit high cosine similarity, suggesting that the layer makes minimal transformations to the hidden states and is therefore less important. We plot the BI scores of a single layer and the PPL after removing it separately, as shown in the Figure 3. The results demonstrate a positive correlation between the BI score and the importance of a layer.

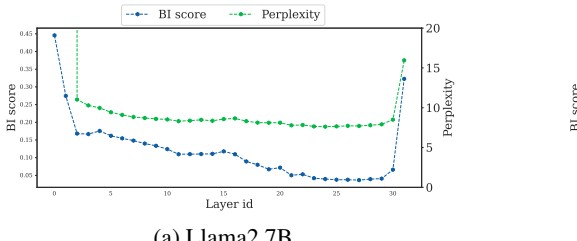 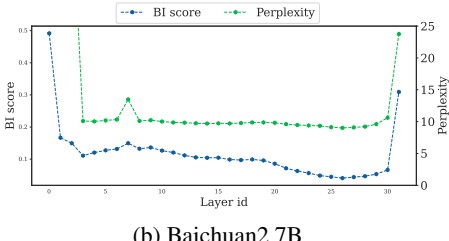

|  |  |
|:-:|:-:|
| (a) Llama2 7B | (b) Baichuan2 7B |

Figure 3: The BI score of a layer and the PPL after removing the layer.

## 3.2 LAYER REMOVAL

Our goal is to obtain a pruned model that remains as close as possible to the original model. Since an LLM functions as a series of transformations applied to hidden states across its layers and we can determine the importance of each layer, we propose a straightforward pruning method: layer removal, which we refer to as ShortGPT. We delete certain layers in LLMs based on BI score. First of all, we construct a calibration set, which is a set of unlabelled text samples such as PG19 (Rae et al., 2019). Then we collect the hidden states of each layer during inference on these samples. Next, we calculate the BI score based on the collected hidden states. Finally, we sort layers in ascending order according to the BI, and delete the layers with the lower BI score. The number of layers to be deleted can vary to trade off the speed and performance. The details of our layer removal setting can be found in Appendix D.

## 4 EXPERIMENTS

### 4.1 EXPERIMENTAL SETUP

**Models.** To validate the effectiveness of our method, we conducted experiments on existing popular open-source language models, including Llama2-7B (Touvron et al., 2023), Llama2-13B, Baichuan2-7B, and Baichuan2-13B. They are all large language models based on the decoder-only Transformer architecture. LLaMA 2 was trained on more than 2 trillion tokens. Baichuan-series was mainly trained in Chinese and its 13-Billion model replaced the RoPE (Su et al., 2024) positional embedding with ALiBi (Press et al., 2021).

**Benchmarks.** In order to comprehensively evaluate the changes in the ability of large language models before and after pruning, we conducted comprehensive evaluation from five aspect: **Reasoning**: CMNLI (Li et al., 2024), HellaSwag (HeSw) (Zellers et al., 2019), PIQA (Bisk et al., 2020). **Language**: CHID (Zheng et al., 2019), WSC (Levesque et al., 2012). **Knowledge**: CommonSenseQA (CoQA) (Reddy et al., 2019), BoolQ (Clark et al., 2019). **Examination**: MMLU (Hendrycks et al., 2020), CMMLU (Li et al., 2024). **Understanding**: Race-High/Middle (H/M) (Lai et al., 2017), XSum (Hasan et al., 2021), C3 (Sun et al., 2020) and PG19 (Rae et al., 2019). For more details, please refer to Appendix G

**Baselines.** To evaluate the effectiveness of our method, we compared several structured pruning methods for large language models, including:

**1) LLMPru** (Ma et al., 2024), which adopts structural pruning that selectively removes non-critical coupled structures based on gradient information, maximally preserving the majority of the LLM's functionality. LLMPru. applies post training to the pruned model, but for fair comparison, we do not apply post training to it.

**2) SliceGPT** (Ashkboos et al., 2024), which is a post-training sparsification scheme that replaces each weight matrix with a smaller matrix, reducing the embedding dimension of the network. Specifically, they applied PCA to the hidden representation from shallow to deep layers, and incorporated the dimension reduction matrix into existing network parameters.

Table 2: Comparison of pruning methods on multiple natural language benchmarks. The results of LLMPrun., SliceGPT and LaCo are reported from LaCo.

| LLM | Method | Ratio | Benchmarks | | | | | | | | | | | | | Ave. | Per. |
|---|---|---|---|---|---|---|---|---|---|---|---|---|---|---|---|---|---|
| | | | CMNLI | HeSw | PIQA | CHID | WSC | CoQA | BoolQ | Race-H | Race-M | XSum | C3 | MMLU | CMMLU | | |
| Llama2-7B | Dense | 0.00% | 32.99 | 71.26 | 77.91 | 41.66 | 50.00 | 64.62 | 71.62 | 35.71 | 34.19 | 19.40 | 43.56 | 45.39 | 32.92 | 47.78 | 100.00 |
| | LLMPrun. | 27.0% | **34.33** | 56.46 | 71.22 | 25.25 | 36.54 | 42.51 | 55.20 | 22.56 | 22.35 | 11.51 | 25.64 | 23.33 | 25.25 | 34.78 | 72.79 |
| | SliceGPT | 26.4% | 31.70 | 50.27 | 66.21 | 20.79 | 36.54 | 41.36 | 38.32 | 21.07 | 21.66 | 4.89 | **39.78** | 28.92 | 25.37 | 32.84 | 68.73 |
| | LaCo | 27.1% | 34.43 | 55.69 | 69.80 | **36.14** | 40.38 | 45.70 | 64.07 | 22.61 | 23.61 | **15.64** | 39.67 | 26.45 | 25.24 | 38.41 | 80.39 |
| | ShortGPT | 27.1% | 32.95 | 53.02 | 66.43 | 24.68 | **52.46** | 47.99 | 74.71 | 32.25 | **35.17** | 0.67 | 39.62 | **43.96** | **32.25** | 41.24 | 86.31 |
| Llama2-13B | Dense | 0.00% | 32.99 | 74.78 | 79.71 | 47.35 | 50.00 | 66.91 | 82.39 | 57.95 | 60.38 | 23.45 | 47.51 | 55.00 | 38.40 | 55.14 | 100.00 |
| | LLMPrun. | 24.4% | **33.03** | 67.76 | **76.66** | 35.64 | 40.38 | 50.86 | 56.42 | 22.47 | 22.08 | **19.17** | 32.33 | 25.21 | 24.71 | 38.97 | 70.67 |
| | SliceGPT | 23.6% | 29.82 | 55.71 | 69.04 | 19.31 | 36.54 | 47.26 | 37.86 | 23.41 | 24.03 | 5.27 | 41.92 | 37.14 | 25.79 | 34.85 | 63.20 |
| | LaCo | 24.6% | 32.86 | 64.39 | 63.20 | **40.10** | **52.88** | 52.66 | **63.98** | 54.49 | 56.55 | 14.45 | 44.93 | 45.93 | 32.62 | 47.62 | 86.36 |
| | ShortGPT | 24.6% | 33.00 | 66.64 | 73.45 | 36.61 | 50.00 | **58.64** | 62.48 | **58.35** | 60.17 | 17.59 | **46.90** | **54.69** | **38.38** | 50.53 | **91.64** |
| Baichuan2-7B | Dense | 0.00% | 33.37 | 67.56 | 76.17 | 85.56 | 50.00 | 63.14 | 74.10 | 52.63 | 51.04 | 20.82 | 64.55 | 53.87 | 56.95 | 57.67 | 100.00 |
| | LLMPrun. | 24.2% | 32.28 | 53.66 | **71.82** | 69.80 | **53.85** | 47.83 | 61.19 | 21.96 | 22.28 | **15.98** | 41.64 | 24.93 | 25.69 | 41.76 | 72.41 |
| | SliceGPT | 22.2% | 32.07 | 25.29 | 50.33 | 14.85 | 36.54 | 19.57 | 39.30 | 23.53 | 22.49 | 0.00 | 26.58 | 25.18 | 25.25 | 26.23 | 45.48 |
| | LaCo | 24.2% | 33.00 | 52.28 | 68.50 | **76.24** | 42.31 | 47.26 | 56.15 | 28.99 | 27.72 | 12.03 | 50.85 | 31.53 | 31.24 | 42.93 | 74.44 |
| | ShortGPT | 24.2% | **33.30** | 56.96 | 67.68 | 65.63 | 50.00 | 46.70 | **67.83** | 53.26 | 46.76 | 0.04 | 56.33 | 45.77 | **47.87** | 49.08 | 85.10 |
| Baichuan2-13B | Dense | 0.00% | 33.21 | 71.10 | 78.07 | 86.51 | 50.00 | 65.6 | 77.89 | 67.27 | 68.94 | 25.02 | 65.64 | 59.50 | 61.30 | 62.31 | 100.00 |
| | LLMPrun. | 24.3% | **33.80** | 53.57 | **71.82** | 72.77 | 37.50 | 38.82 | 56.54 | 21.17 | 21.61 | 13.67 | 39.89 | 23.19 | 25.18 | 39.20 | 62.91 |
| | SliceGPT | 22.8% | 32.07 | 25.85 | 51.03 | 10.40 | 36.54 | 18.02 | 37.83 | 21.56 | 21.52 | 0.00 | 24.99 | 22.95 | 25.26 | 25.23 | 40.49 |
| | LaCo | 24.7% | 33.03 | **60.71** | 68.88 | 76.73 | 44.23 | **55.45** | 62.35 | 56.92 | 57.80 | 12.32 | **61.10** | 51.35 | 53.65 | 53.43 | 85.75 |
| | ShortGPT | 24.7% | 32.81 | 60.55 | **71.60** | **80.17** | 47.13 | 54.30 | **62.54** | 55.77 | 56.41 | **15.14** | 60.16 | **52.11** | **58.86** | 54.43 | 87.35 |

**3) LaCo** (Yang et al., 2024), which is a pruning method for large language models based on reducing layers. LaCo gradually merges similar layers from deep to shallow and sets a threshold to avoid continuously merging too many layers.

For our evaluation, we use PG19 for layer importance and perplexity calculation. The models, baselines and evaluate benchmarks is the same as LaCo.

## 4.2 MAIN RESULTS

To validate the efficacy of our proposed method, we conducted comparative experiments against baseline techniques commonly employed in large language model evaluation. Considering the current structured pruning methods generally reduce parameters by no more than 30%, we performed experiments with approximately 1/4 of the parameters pruned. The experimental results are presented in Table 2. Additional experiments exploring different parameter reduction proportions will be discussed in the subsequent section.

The results demonstrate that the performance of the model pruned by our method significantly surpasses that of the baseline methods, maintaining most of the large language model's capabilities. Furthermore, we note that the approach of reducing the number of layers (ShortGPT/LaCo) outperforms the method of reducing the embedding dimensions (LLMPru./SliceGPT), implying that the model exhibits more redundancy in depth than in width. Further experimental analysis will be presented in the ensuing section.

In Table 2, we fully adopted the benchmark, model, and pruning ratio in the LaCo paper. In order to make a more fair comparison with LLMprun. and SliceGPT, we compared them with the same benchmark, model, and pruning ratio in their original paper. The experimental results are shown in Appendix C. Consistent with our findings in Table 2, these experiments further demonstrate the significant layer redundancy present in existing large language models, and ShortGPT achieves superior performance compared to other pruning methods.

The results show that coarse-grained pruning methods, such as removing entire layers, often outperform fine-grained approaches like Slice GPT or LLM Pruner. We speculate that the reason is that the large language model is actually very robust, as shown in Figure 1, removing any deep layer individually actually has very little impact on the final output, which means it is difficult to define the importance of a finer grained module and perform pruning.

## 4.3 VARYING METRIC AND PRUNING RATIO

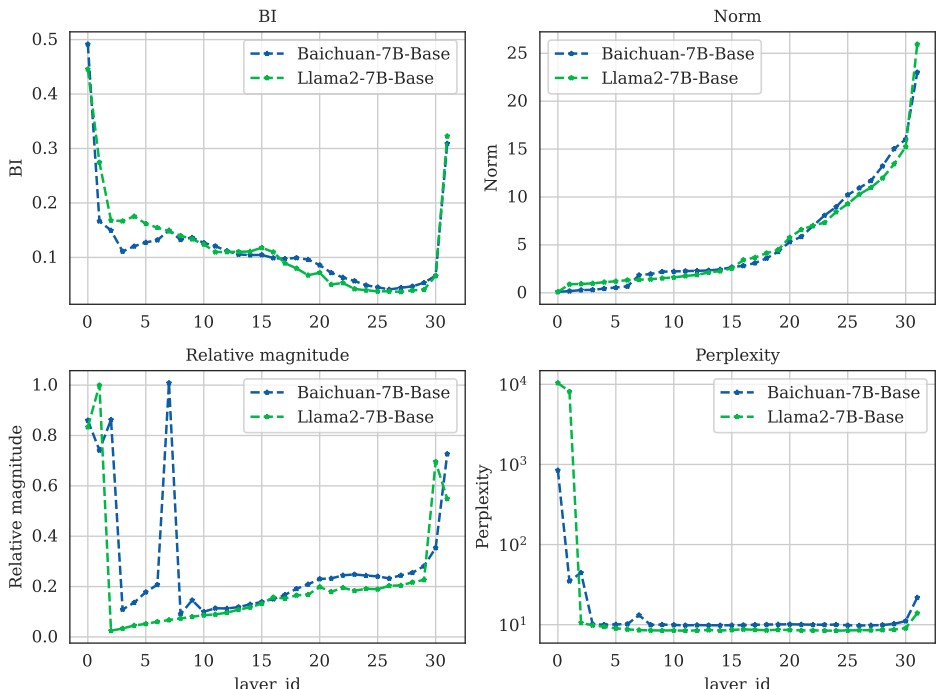

Figure 4: Comparison of different importance metrics. Perplexity is calculated by removing each single layer, other metrics is calculated by hidden states of each layer.

The core principle of our method is to rank layers by their importance and remove the less significant ones. The choice of importance metric significantly influences the outcome. In this section, we define and compare several different importance metrics:

- **Sequential**: The importance is directly proportional to the sequence order, with shallower layers being less important. This can be implemented by assigning the negative value of each layer's index as its importance metric.

- **Norm/Reverse-order**: This metric posits that importance is inversely proportional to the sequence order. It assigns higher importance scores to the shallower layers. This method gives the same order as measuring importance by hidden states norm as Figure 4 shows.

- **Relative Magnitude**: Proposed in Samragh et al. (2023), this metric assumes layers with larger $||\frac{f(x)}{x+f(x)}||$ are of higher importance, where $f$ is the layer transformation function.

- **BI**: we calculate the BI score mentioned in Section 3.1 as importance metric.

Figure 4 demonstrates the different metrics. We observe that shallower layers in the LLM network are more crucial than deeper ones. Figure 5 shows the results of removing layers by different metrics, demonstrating that Our proposed BI outperforms other metrics. The method of Relative Magnitude is highly competitive, indicating that relative values can also reflect the importance to some extent. It is worth noting that the hidden states norm seems to be a good metric when only considering the MMLU benchmark, but the perplexity is relatively poor.

As a pruning method, we further validated the effects of different pruning ratios on model performance. Experiments were conducted on the Llama2 and Baichuan2 models, observing the Perplexity and MMLU. The results for Llama2, as shown in Figure 5, demonstrate that the model's performance generally declines as the pruning ratio increases. However, we observe a notable phenomenon: the MMLU score exhibits a sharp drop at a specific layer. This sudden decrease suggests the presence of certain critical layers within the network that play a particularly important role in

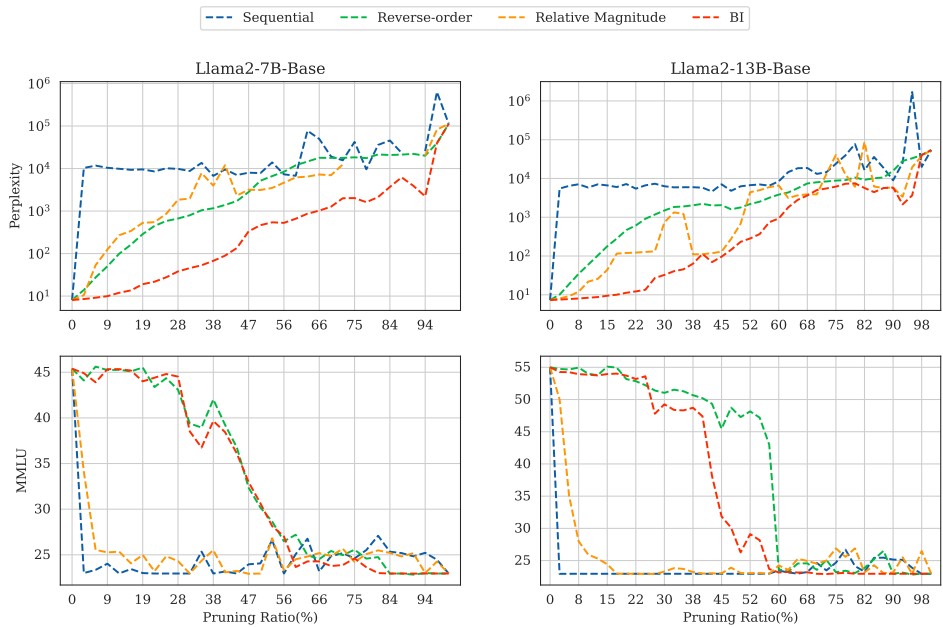

Figure 5: Performance of MMLU and perplexity when we prune by different metrics, with increasing pruning ratio. We can see that as the pruning ratio increases, the performance of the model declines.

Table 3: ShortGPT pruning on RWKV and Mamba.

| Model | Pruning ratio | CMNLI | HeSw | PIQA | CHID | WSC | CoQA | BoolQ | Race-H | Race-M | XSum | C3 | MMLU | CMMLU | Ave. | Per. |
|---|---|---|---|---|---|---|---|---|---|---|---|---|---|---|---|---|
| Mamba2.8B | 0% | 35.97 | 61.84 | 75.52 | 35.56 | 49.69 | 56.35 | 60.67 | 24.9 | 25.3 | 15.03 | 42.08 | 26.29 | 25.32 | 41.12 | 100.00 |
| | 10.9% | 32.95 | 59.71 | 73.01 | 32.52 | 49.28 | 52.66 | 51.41 | 24.27 | 25.21 | 14.95 | 41.1 | 26.01 | 25.00 | 39.08 | 95.04 |
| | 20.3% | 31.29 | 55.69 | 69.64 | 29.12 | 48.36 | 48.32 | 62.2 | 23.61 | 23.61 | 14.71 | 41.59 | 25.69 | 25.37 | 38.36 | 93.29 |
| | 25% | 29.96 | 52.38 | 68.77 | 26.02 | 48.26 | 44.96 | 62.2 | 23.67 | 23.26 | 14.00 | 40.71 | 24.32 | 24.89 | 37.18 | 90.42 |
| | 31.3% | 28.25 | 47.02 | 64.91 | 21.38 | 49.69 | 44.96 | 62.17 | 21.87 | 22.77 | 13.77 | 40.44 | 24.48 | 24.77 | 35.59 | 86.55 |
| RWKV7B | 0% | 32.07 | 65.98 | 77.09 | 85.36 | 50.00 | 62.65 | 62.72 | 38.56 | 45.47 | 16.5 | 57.97 | 31.85 | 28.54 | 50.37 | 100.00 |
| | 9.4% | 32.6 | 56.41 | 73.94 | 78.12 | 50.00 | 49.55 | 62.35 | 25.9 | 25.77 | 9.57 | 54.68 | 27.29 | 25.03 | 43.94 | 87.23 |
| | 18.8% | 32.11 | 49.47 | 71.55 | 65.63 | 50.00 | 40.54 | 61.19 | 22.04 | 23.75 | 8.13 | 49.15 | 26.35 | 25 | 40.38 | 80.17 |
| | 25% | 32.41 | 39.73 | 65.13 | 52.6 | 50.00 | 29.65 | 60.92 | 22.56 | 21.59 | 12.02 | 41.86 | 25.52 | 25.08 | 36.85 | 73.16 |
| | 28.1% | 33.11 | 32.22 | 60.01 | 32.47 | 50.1 | 28.34 | 60.85 | 22.27 | 21.31 | 10.43 | 37.81 | 25.64 | 25.15 | 33.82 | 67.14 |

maintaining performance. Similar patterns are observed in the Baichuan2 model, as illustrated in Appendix B.

## 4.4 REDUNDANCY ON NON-TRANSFORMER LLM

To determine whether the observed depth redundancy is specific to the Transformer architecture, we extended our investigation to include two popular non-Transformer models, RWKV-7B[1] (Peng et al., 2023) and Mamba-2.8B [2] (Gu & Dao, 2023). Our experiments revealed that these models also exhibit resilience to layer removal, maintaining performance despite the elimination of certain layers. This finding suggests that the redundancy phenomenon may not be unique to Transformer-based models, but rather a common characteristic across current large language models. Table 3 shows that our method is applicable and effective for both Mamba and RWKV models, suggesting that the redundancy is universal across current LLMs. However, it is worth noting that the RWKV model appears less redundant than Mamba and Transformer models, which warrants further investigation.

---

[1] We use rwkv-v5-world-7B from https://huggingface.co/RWKV/v5-Eagle-7B-HF
[2] We take the model from https://huggingface.co/state-spaces/mamba-2.8b-hf

Table 4: Layer removal results on Llama2-7B-Base-GPTQ.

| Model | Ratio/Layer | Perplexity | MMLU | Throughput (speed up) |
|---|---|---|---|---|
| Baseline | 0%/32 | 8.03 | 43.17 | 4331.23 Token/s (1.00x) |
| ShortGPT | 3.1%/31 | 8.37 | 42.88 | 4399.31 Token/s (1.02x) |
| | 9.4%/29 | 9.44 | 42.31 | 4602.26 Token/s (1.06x) |
| | 12.5%/28 | 10.24 | 41.62 | 4680.68 Token/s (1.08x) |
| | 15.6%/27 | 11.42 | 43.17 | 4756.94 Token/s (1.10x) |
| | 25.0%/24 | 22.29 | 41.68 | 5045.59 Token/s (1.16x) |
| | 27.1%/23 | 40.78 | 43.35 | 5146.99 Token/s (1.19x) |

Table 5: Performance comparison of different methods

| Method | MMLU | CMMLU |
|---|---|---|
| Llama2-7B-Baseline | 45.4 | 32.9 |
| 4-bit quantization | 44.9 | 32.5 |
| Layer removal (27.1%) | 44.0 | 32.3 |
| 4-bit quantization then layer removal | 42.4 | 31.0 |
| Layer removal then 4-bit quantization | 41.2 | 30.5 |

## 4.5 ORTHOGONAL TO QUANTIZATION

In this section, we show that our method is orthogonal to quantization methods. We apply our method to Llama2-7B [3] quantized by GPTQ algorithm. Table 4 shows that our method is compatible with the quantization-like method. In addition, we compared the performance of applying pruning before quantization [4]. The results shown in the Table 5 further indicates that quantization and ShortGPT are orthogonal operations.

## 4.6 POST TRAINING TO RESTORE PERFORMANCE

To mitigate the performance loss resulting from layer removal, we explored post-training strategies inspired by Chen et al. (2024). Our approach comprised two key steps: 1)Replacement: We substituted the removed layers with lightweight Multi-Layer Perceptron (MLP) modules. 2)Retraining: We subsequently retrained the modified model. The results in Table 6 demonstrate the potential of post-train in recover performance loss. Appendix F list the training details.

Table 6: Post-train Llama2-7B to restore performance.

| Method | Avg. | Ratio | CMNLI | HeSw | PIQA | CHID | WSC | CoQA | BoolQ | Race-H | Race-M | XSum | C3 | MMLU | CMMLU |
|---|---|---|---|---|---|---|---|---|---|---|---|---|---|---|---|
| Dense | 47.78 | 0% | 32.99 | 71.26 | 77.91 | 41.66 | 50.00 | 64.62 | 71.62 | 35.71 | 34.19 | 19.40 | 43.56 | 45.39 | 32.92 |
| ShortGPT | 41.22 | 27.1% | 32.95 | 53.02 | 66.43 | 24.68 | 52.46 | 47.99 | 74.41 | 32.25 | 35.17 | 0.67 | 39.62 | 43.96 | 32.25 |
| ShortGPT+post-train | 43.16 | 24.0% | 32.99 | 54.83 | 68.12 | 31.82 | 51.37 | 58.32 | 72.36 | 34.18 | 34.68 | 4.89 | 40.37 | 44.47 | 32.73 |

## 5 LIMITATION

Although our method demonstrates strong competitiveness compared to current pruning methods, there are some phenomena that have not been explained. Our experiments reveal that the negative effect of layer removal is more significant on generative tasks compared to multiple-choice tasks. When we remove 25% layers from Llama2-7B or Baichuan2-7B, the performance in generative

---

[3]We take the model from https://huggingface.co/TheBloke/Llama-2-7B-GPTQ

[4]We use GPTQ algorithm for quantization from https://github.com/AutoGPTQ/AutoGPTQ

tasks such as XSum and C3 deceases to nearly zero, although the performance decline was not as significant on the larger model of the 13B. We speculate that compared to multiple-choice tasks, generative tasks face the problem of accumulated errors and large model is more robust than small one. The reasons behind it still need to be explored. The post-training techniques discussed in Section 4.6 have the potential to mitigate this issue and warrant further exploration.

# 6 RELATED WORKS

To reduce the inference cost of large language models and increase their practical applications, there have been many recent works on compressing models, which can be classified into two categories: model pruning and quantization. Besides, there are some works aim to study the redundancy of model which is essential for compressing models.

**Model pruning:** model pruning (LeCun et al., 1989; Han et al., 2015) is a classic and effective method of reducing model redundancy modules to compress models. The model pruning methods mainly include unstructured pruning and structured pruning. The unstructured pruning simplifies an LLM by removing specific parameters without considering its internal structure, such as SparseGPT (Frantar & Alistarh, 2023) and LoRAPrune (Zhang et al., 2023). However, this method disregards the overall LLM structure, resulting in an irregular sparse model composition. Another more practical approach is structured pruning, GUM(Syed et al., 2023) makes an analysis of several structured pruning methods for decoder-only LLMs. LLM-Pruner (Ma et al., 2024) selectively removes non-critical structures according to gradient information. ShearedLLaMA (Xia et al., 2023) employs targeted structured pruning and dynamic batch loading. LaCo (Yang et al., 2024) used layer merging to compress the model. Compared to the previous method, our method is a simple and efficient structured pruning method.

**Quantization:** quantization (Liu et al., 2021; Gholami et al., 2022; Dettmers et al., 2022; 2024) is a widely accepted technique in the field of model compression, which can significantly save the storage and computational costs of deep learning models. Traditional models are generally stored as floating-point numbers, but quantization converts them into integers or other discrete forms. LUT-GEMM (Park et al., 2022) quantifies only weights and optimizes matrix multiplication in LLM using BCQ format. SPQR (Dettmers et al., 2023) identifies and isolates abnormal weights, stores them with higher accuracy, and compresses all other weights into 3-4 bits. Our model pruning method and quantization method are orthogonal, which means quantification based on our pruned model can further compress the model.

**Model redundancy:** researchers have long noticed the significant redundancy in nonlinear models (Catchpole & Morgan, 1997). In recent years, the transformer model architecture has been widely applied, and researchers have also studied its redundancy. In (Bian et al., 2021), researchers analyzed redundancy in attention mechanisms, in which clear and similar redundancy patterns (cluster structure) are observed among attention heads. In (Dalvi et al., 2020), researchers dissect two pre-trained models, BERT (Devlin et al., 2018) and XLNet (Yang et al., 2019), studying how much redundancy they exhibit at a representation level and a more fine-grained neuron-level. However, the redundancy in current large language models based on decoder-only structures still needs to be explored.

# 7 CONCLUSION

In this work, we uncovered the significant layer-wise redundancy of LLMs, Our research demonstrates that certain layers contribute minimally to overall network functionality and can be removed without substantially compromising model performance. Based on our observation, We introduce Block influence to quantify the importance of each layer and propose a simple and straightforward pruning method: layer removal. Our experiments demonstrates that it is possible to maintain up to approximately 90% of a LLM's performance while reducing the model's parameter amount and computational requirements by approximately 25%. Besides, our method is orthogonal to quantization methods and can be further improved by continual training. We hope that our work can provide some insight for future model compression techniques. Moreover, our work suggests potential avenues for improving the efficiency of model training by reducing inherent redundancy in the future.

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

# A MATHEMATICAL EXPLANATION FOR WHY PRE-NORM BRINGS HIGH SIMILARITY

We provide a simple explanation here about how pre-norm leads to high deep similarity in this section, here we adopt RMSNorm (Zhang & Sennrich, 2019) for convenient, which is also the popular pre-norm used in many recent LLMs, such as Llama and Mamba.

**Lemma 1.** *(Xiong et al., 2020) At initialization, for the Pre-LN Transformer, $(1 + \frac{L}{2})d \leq \mathbb{E}(||x_{L,i}||_2^2) \leq (1 + \frac{3L}{2})d$ for all $L > 0$ and $i$. Expectations are taken over the input and the randomness of initialization, where the hidden state of $L^{th}$ layer is $x_L$.*

From Lemma 1, the hidden state of the pre-norm model will continuously increase as the number of layers increases. And under the assumption of each component of $x_l$ has a mean of 0, we can obtain $||x_L|| = \Theta(\sqrt{L})$.

Then we consider $x_{L+1} = x_L + f_L(x_L, \theta_L)$, where $f_L$ is a operation such as Attention or MLP, $\theta_L$ is learnable parameters. Then $f_L(x_L, \theta_L) = O(1)$ respect to $L$, for Attention as example, $||f_L(x_L, \theta_L)|| = ||(softmax(Q^T K)X_L/||X_L|| \cdot (\sigma_{rms}))W_v W_q|| = O(||\sigma_{rms}||||W_v||||W_o||) = O(1)$ respect to $L$.

Then we can get:

$$cos\ similarity(X_{L+1}, X_L) = \frac{x_{L+1}x_L}{||x_{L+1}||||x_L||} = \frac{||x_L||^2}{||x_{L+1}||||x_L||} + \frac{f_L(x_L, \theta)x_L}{||x_{L+1}||||x_L||} \tag{2}$$

$$\geq \frac{||x_L||^2}{||x_{L+1}||||x_L||} - \frac{||f_L(x_L, \theta)||||x_L||}{||x_{L+1}||||x_L||} \tag{3}$$

$$= \frac{||x_L||}{||x_{L+1}||} - \frac{||f_L(x_L, \theta)||}{x_{L+1}} = \Theta(\sqrt{\frac{L}{L+1}}) - O(\sqrt{\frac{1}{L+1}}) \tag{4}$$

This means that as the number of layers $L$ increases, the similarity between the input and output of the layer will be high. This means that the role of $f_L$ may be relatively small, and removing it from the network may have a relatively small impact to the model.

Although the above theoretical analysis is only for randomly initialized models, this phenomenon that deep layer has similar input and output exists in both our own trained models shown in Figure 2 and existing models in Figure 4.

# B LAYER REMOVAL ON BAICHUAN2-SERIES MODEL

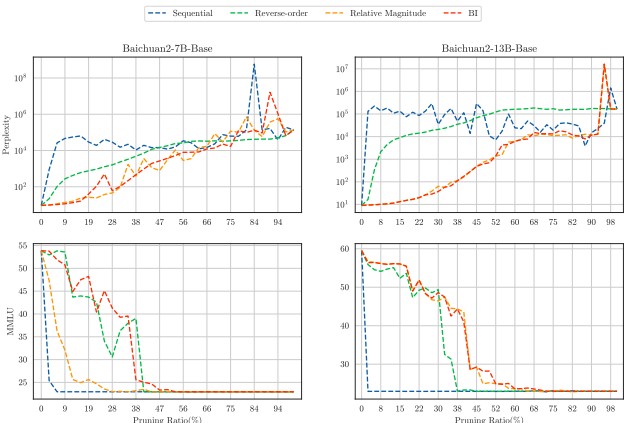

Figure 6: Pruning by different metrics on Baichuan2-series model.

## C  A FAIR COMPARISON WITH SLICEGPT AND LLMPRUN.

In Table 2, we fully adopted the benchmark, model, and pruning ratio in the LaCo's paper. For a fair comparison with LLM pruner and SliceGPT, we do the same experiments in the original paper of LLM pruner and SliceGPT. The results is provided in Table 7 and Table 8. We take **the same benchmarks, models and pruning ratio** as the corresponding original paper. The results demonstrate that our method is highly competitive.

Table 7: Comparison between ShortGPT and LLM-pruner. The Table is corresponding to the Table 1 of LLM pruner(Zhang et al., 2023).

| Model | Pruning ratio | Method | BoolQ | PIQA | Hellaswag | Winogrande | Arc-e | Arc-c | OBQA | Avg. |
|---|---|---|---|---|---|---|---|---|---|---|
| | Ratio=0% | Baseline | 73.18 | 78.35 | 72.99 | 67.01 | 67.45 | 41.38 | 42.4 | 63.25 |
| Llama-7B | Ratio=20% | LLM-pruner | 59.39 | 75.57 | 65.34 | 61.33 | 59.18 | 37.12 | 39.80 | 56.82 |
| | Ratio=21.9 % | ShortGPT | 68.26 | 72.28 | 61.7 | 63.77 | 60.22 | 39 | 41.6 | 58.12 |
| | Ratio=0% | Baseline | 68.47 | 78.89 | 76.24 | 70.09 | 74.58 | 44.54 | 42.00 | 64.97 |
| Llama-13B | Ratio=20% | LLM-pruner | 67.68 | 77.15 | 73.41 | 65.11 | 68.35 | 38.4 | 42.4 | 61.79 |
| | Ratio=20% | ShortGPT | 68.41 | 76.36 | 72.9 | 67.4 | 68.62 | 39.2 | 41 | 61.98 |

Table 8: Comparison between ShortGPT and SliceGPT. The Table is corresponding to the Table 7 of SliceGPT(Ashkboos et al., 2024).

| Model | Pruning ratio | Method | PIQA | Hellaswag | Winogrande | Arc-e | Arc-c | Avg. |
|---|---|---|---|---|---|---|---|---|
| | 0% | Baseline | 79.11 | 75.99 | 69.06 | 74.58 | 46.25 | 69 |
| Llama-2-7B | 20% | SliceGPT | 71.87 | 58.1 | 63.04 | 69.87 | 43.09 | 63.45 |
| | 25% | SliceGPT | 68.55 | 58.1 | 62.04 | 57.46 | 35.07 | 56.15 |
| | 30% | SliceGPT | 66.1 | 52.69 | 56.82 | 35.07 | 56.82 | 56.15 |
| | 21.9% | ShortGPT | 72.76 | 66.39 | 66.27 | 59.39 | 39.85 | 60.93 |
| | 25% | ShortGPT | 70.53 | 62.68 | 64.7 | 58.39 | 39.51 | 59.16 |
| | 31.6% | ShortGPT | 67.87 | 62.19 | 64.38 | 56.57 | 40.86 | 58.37 |
| | 0% | Baseline | 80.47 | 79.39 | 72.22 | 77.48 | 49.23 | 71.76 |
| Llama-2-13B | 20% | SliceGPT | 71.87 | 69.38 | 63.04 | 69.87 | 43.09 | 63.45 |
| | 25% | SliceGPT | 68.55 | 67.48 | 58.1 | 62.5 | 37.88 | 58.9 |
| | 30% | SliceGPT | 66.1 | 65.11 | 52.69 | 56.82 | 35.07 | 55.16 |
| | 20% | ShortGPT | 76.95 | 74.67 | 71.14 | 69.56 | 45.63 | 67.59 |
| | 25% | ShortGPT | 74.39 | 71.65 | 70.98 | 67.09 | 43.93 | 65.61 |
| | 30% | ShortGPT | 72.11 | 71.93 | 67.19 | 61.09 | 40.88 | 62.64 |
| | 0% | Baseline | 82.7 | 83.84 | 77.98 | 80.98 | 57.34 | 76.57 |
| Llama-2-70B | 20% | SliceGPT | 76.61 | 72.98 | 74.92 | 80.51 | 55.2 | 72.34 |
| | 25% | SliceGPT | 74.92 | 68.74 | 74.92 | 77.9 | 51.71 | 69.75 |
| | 30% | SliceGPT | 72.31 | 63.69 | 73.4 | 51.71 | 47.61 | 66.11 |
| | 20% | ShortGPT | 76.02 | 78.87 | 71.69 | 76.02 | 52.95 | 71.68 |
| | 25% | ShortGPT | 73.2 | 76.72 | 71.85 | 73.2 | 49.9 | 69.79 |
| | 30% | ShortGPT | 74.44 | 75.31 | 72.33 | 74.44 | 49.22 | 69.4 |

## D  DETAILED STRATEGIES FOR LAYER REMOVAL

We list the details of different layer removal strategies in Table 10. The concrete removed layers by ShortGPT in Table 2 are listed in Table 9

Table 9: Setup of Removed Layers for Benchmark Models.

| Model | Removed Layers |
|---|---|
| Llama-2-7B | 27, 26, 25, 28, 24, 29, 23, 21, 22 |
| Llama-2-13B | 33, 31, 32, 30, 29, 34, 28, 35, 27, 26 |
| Baichuan-2-7B | 26, 27, 25, 28, 24, 29, 23, 22, 30 |
| Baichuan-2-13B | 32, 31, 33, 30, 34, 29, 28, 35, 27, 26 |

## E  SETUP FOR TRAINING POST-NORM MODEL AND PRE-NORM MODEL

We have listed the specific training settings for pre norm and post norm in Table 11.

Table 10: Strategies for Layer Removal in Models.

| Strategy | Description |
| --- | --- |
| Sequential | Layers are removed sequentially from the beginning of the model. The process starts with layer 0 and progressively includes more layers for removal (e.g., {0}, {0, 1}, . . . ). |
| Reverse-order | This strategy involves starting from the model's final layer and progressively removing layers in reverse order (e.g., {-1}, {-1, -2}, . . . ). |
| Relative Magnitude | Layers are removed in ascending order based on their Relative Magnitude values. The removal process accumulates layers from those with the smallest to the largest values, mirroring the sequential strategy's accumulation method. |
| BI (Block Influence) | Follows a similar accumulation approach as the Sequential strategy, but layers are ordered and removed according to their BI values, starting from the lowest and moving to the highest. |

Table 11: Training Parameters.

| Parameter | Value |
| --- | --- |
| Global Batch Size | 2048 |
| Sequence length | 4096 |
| Precision | bf16 |
| Learning Rate Scheduler | cosine |
| Max Learning Rate | 4e-4 |
| Min Learning Rate | 5e-5 |
| Warm-up steps | 3000 |
| Training Tokens | 200B |
| Weight Decay | 0.1 |
| Adam Beta1 | 0.9 |
| Adam Beta2 | 0.98 |
| Gradient Clip | 1.0 |
| Tokenizer | Llama2 |
| Layers | 32 |
| Hidden state | 2048 |
| Attention heads | 32 |
| Head dim | 64 |
| FFN size | 5504 |
| Activation function | Silu |

## F   POST-TRAINING SETTINGS

We replace the removed layer with a lightweight gated MLP layer with hidden size = 2048. Table 12 show the post training settings.

Table 12: Post-training Parameters.

| Parameter | Value |
| --- | --- |
| Global Batch Size | 2048 |
| Sequence length | 4096 |
| Precision | bf16 |
| Learning Rate Scheduler | cosine |
| Max Learning Rate | 2e-5 |
| Min Learning Rate | 1e-5 |
| Warm-up steps | 3000 |
| Training Tokens | 50B |
| Weight Decay | 0.1 |
| Adam Beta1 | 0.9 |
| Adam Beta2 | 0.98 |
| Gradient Clip | 1.0 |

## G Evaluation Benchmarks

In order to comprehensively evaluate the changes in the ability of large language models before and after pruning, we conducted evaluations on the most commonly used Benchmark MMLU Hendrycks et al. (2020), CMMLU Li et al. (2024) for evaluating large models. In addition, we also followed LaCo Yang et al. (2024) to evaluate a wider dataset.

**MMLU** Hendrycks et al. (2020) is a benchmark aimed at measuring the knowledge acquired during pre-training by specifically evaluating models in zero-shot and few-shot settings. This makes benchmarks more challenging and similar to the way we evaluate humans. This benchmark covers 57 subjects including STEM, humanities, social sciences, etc. Its difficulty ranges from beginner to advanced professional level, and it tests world knowledge and problem-solving ability.

**CMMLU** Li et al. (2024) is a comprehensive Chinese language assessment dataset designed specifically to evaluate LLM's advanced knowledge and reasoning abilities in the context of Chinese language and culture. CMMLU covers 67 topics, from elementary school to university or professional level. Including natural sciences, as well as humanities and social sciences, it also includes many contents with Chinese characteristics.

**CMNLI** Xu et al. (2020) is part of the Chinese language understanding assessment benchmark. It consists of two parts: XNLI and MNLI. **HellaSwag (HeSw)** Zellers et al. (2019) is a challenging dataset for evaluating commonsense NLI that is especially hard for state-of-the-art models, though its questions are trivial for humans. **PIQA** Bisk et al. (2020) is a multi-choice question and answer dataset that focuses on daily scenarios. This dataset explores the model's grasp of the laws of the real physical world through daily scenarios. **CHID** Zheng et al. (2019) is an idiom cloze test dataset that mainly focuses on the selection of candidate words and the representation of idioms. **CoQA** Reddy et al. (2019) is a large-scale dataset used for conversational question-answering tasks, containing over 127000 questions and their corresponding answers. **BoolQ** Clark et al. (2019) is a question-answer dataset containing 15942 examples of yes/no questions. These problems occur naturally - they are generated in an environment that is silent and unconstrained. **Race** Lai et al. (2017) is a large-scale reading comprehension dataset collected from English examinations in China, which are designed for middle school and high school students. **XSum**Hasan et al. (2021) is used to evaluate abstract single document summarization systems. The goal is to create a short, one-sentence new summary of what the article is about. **C3** Sun et al. (2020) is a machine reading comprehension dataset with multiple choices, consisting of multiple-choice questions, reading materials from Chinese proficiency exams, and ethnic Chinese exams. **PG19** Rae et al. (2019) is a long document dataset from books used to test the effectiveness of language modeling.

## H Hardware Environment

The platform we use to experiment is GPU heterogeneous platform. The hardware of our platform is shown in Table 13

Table 13: Setup of Removed Layers for Benchmark Models.

| Name | Details |
| --- | --- |
| CPU | 2x Intel(R) Xeon(R) Gold 6430 CPU @ 2.1GHz |
| GPU | 8x NVIDIA A100-80GB Tensor Core GPU |

