# OpenReview forum: "ShortGPT: Layers in Large Language Models are More Redundant Than You Expect"
_ICLR.cc/2025/Conference — Submitted to ICLR 2025_

### Official Review · Reviewer_bT5h · 2024-10-31

**Soundness:** 3
**Presentation:** 3
**Contribution:** 2
**Rating:** 5
**Confidence:** 4

**Summary:**

This paper introduces a layer pruning metric named BI and demonstrates its superior performance over previous state-of-the-art pruning methods on Llama2-7B and 13B, Baichuan2-7B and 13B. Besides, ShortGPT can be used together with other quantitative methods for further reduction in parameters and computation.

**Strengths:**

- This paper introduces a new layer pruning method, BI.
- This paper conducts experiments on non-transformer models, RWKV-7B and Mamba-2.8B.
- The paper contains many comparative experiments.

**Weaknesses:**

- Insufficient contribution. ShortGPT is orthogonal to the quantization method, which cannot be regarded as a contributing factor, as other pruning methods can achieve similar results.
- In this paper, PG19 is used for layer importance and perplexity calculation. What if we use other datasets like alpaca-cleaned [1] and mmlu?
- Table 5 can be adjusted to make it more beautiful.
- In eq.1, in fact, BI is a cosine distance, so can other distance formulas also be used as metrics for calculation? Why must it be cosine?

[1] alpaca-cleaned: https://huggingface.co/datasets/yahma/alpaca-cleaned

**Questions:**

See above.

---

> ### Author Response · Authors · 2024-11-24
>
> Thank you for your thorough review and for pointing out weaknesses.
> 1. **Regarding the contribution and orthogonality with quantization methods**: We agree that other pruning methods can also perform quantization. However, advanced pruning methods often rely on complex intermediate computations. The cumulative biases introduced by quantization can significantly affect their pruning results. In contrast, our layer-wise pruning method yields nearly identical results whether quantization is applied before or after pruning, demonstrating their orthogonality. Regarding the contribution of our paper, we emphasize **the layers in large language models could be more redundant than expected and provide theoretical proofs**. This contributes to a deeper understanding of current large models rather than solely focusing on proposing a sota pruning method.
> 2. **On the choice of calibration dataset**: This is an excellent point. We have supplemented our results by choosing MMLU as the calibration dataset for calculating BI values:
> | Model       | PG19 Layers Removed   | MMLU Layers Removed   | MMLU (Calib PG19) | MMLU (Calib MMLU) | MMLU (Full Layers) |
> |-------------|-------------|-------------|-------------|-------------|--------------|
> | LLaMA-2-7B  | ['25', '27', '26', '24', '28', '29', '23', '22', '21'] | ['27', '25', '26', '28', '24', '23', '29', '22', '21'] | 44.5              | 44.5              | 45.9               |
> | LLaMA-2-13B | ['33', '31', '32', '30', '34', '29', '35', '28', '27', '36'] | ['31', '33', '32', '29', '30', '35', '34', '27', '28', '36'] | 48.5              | 48.5              | 55.7               |
>
> In the table, the numerical sequences represent layers sorted by BI from smallest to largest. From the results, we observe that different calibration datasets have minimal impact on the BI calculation, with only slight variations in order due to normal fluctuations—since the BI values of these layers are already very small. Moreover, calculating BI using the MMLU dataset does not provide additional performance improvements on MMLU. This is because layer-based pruning is a coarse-grained method; we measure the expected transformation ability of each layer, which does not bias toward specific MMLU problems. Furthermore, we believe that under the law of large numbers, any semantic dataset will yield similar results, which is one of the reasons we directly chose the PG19 dataset.
>
> 3. On improving Table 5: Thank you for your suggestion regarding the details of the paper. We will enhance the presentation of Table 5 to make it more aesthetically pleasing and clearer for readers.
> 4. **Regarding the use of cosine distance in Equation 1**: This is a profound question. It is proven that norms in finite-dimensional linear spaces are equivalent, so theoretically, any distance metric could be used. However, the Transformer layer is a nonlinear transformation, so we need to carefully consider which distance metric best characterizes BI. We chose the cosine distance based on considerations of directionality. From a finer-grained perspective, the hidden state is followed by the critical attention module. Before computing the softmax, the attention mechanism is a linear transformation, and the softmax does not change the relative magnitudes of the attention scores. Therefore, the direction of the input hidden state is crucial to the attention results. We believe that as long as the direction of the hidden state remains unchanged, the computation in the attention module will not undergo fundamental changes. Consequently, we characterize the transformation capability of the Transformer layer by the change in the direction of the hidden state, using the cosine distance to calculate BI.

---

> > ### Comment · Reviewer_bT5h · 2024-11-27
> >
> > Thank you for your clarification. As you say, any distance metric could be used for calculating the layer importance. So I think the authors need more experiments to demonstrate the effectiveness of cosine distance.

---

> > > ### Author Response · Authors · 2024-11-27
> > >
> > > Thank you for your response. However, I would like to further clarify a point regarding the equivalence of distance metrics. My previous statement about distance metrics being equivalent in finite-dimensional linear spaces may have been misleading, as it is well-known that the entire Transformer layer is a nonlinear transformation. Therefore, we cannot arbitrarily select a distance metric to characterize the layer’s importance. The selection of cosine distance is motivated by our aim to directly capture the transformation capability of the Transformer layer in terms of its directional change. We have also validated this approach through experiments on 13 benchmarks, which have yielded promising results.
> > > Of course, your suggestion is valuable, but we would like to emphasize that  **the main focus of this work is to investigate, both theoretically and experimentally, the redundancy of layers in Large Language Models , rather than exploring pruning metrics**.

---

### Official Review · Reviewer_nyGp · 2024-11-01

**Soundness:** 3
**Presentation:** 2
**Contribution:** 3
**Rating:** 5
**Confidence:** 4

**Summary:**

This paper proposes a metric; Block Influence (BI) which the authors use as a proxy to remove the layers (prune) without affecting (minimal affect) on the model performance. The metric is assumed to capture the redundancy component of the layer, so intuitively, removing the blocks which are most redundant i.e lower BI makes sense.

**Strengths:**

1. The paper is easy to read and clear at most of the places, the flow is good but can be made even better.
2. The conceptual idea of coming up with metric (Block Influence) makes sense intuitively (section 2 and section 3) and it's a simple metric to implement/understand.
3. The experiments clearly demonstrate the effectiveness of the method.

**Weaknesses:**

**Clarifications**
1. From table 9 and other figures, the authors might have to clearly define what a "layer" means because the metric in use is Block Importance (BI), so the normal understanding would be Attention + FFN is a block (the original transformer has 16 blocks etc;). So from my understanding, when layer 5 is removed, I believe the Attention + FFN is replaced by Identity matrix. Is that correct? This definition has to clearly come out at the start.
2. The finding presented in L256-258 is interesting i.e more redundancy in depth compared to width. If there's any previous work, the authors can cite it. By plainly looking at Table 2, this statement cannot be totally true and needs further validation.

**Additional Experiments/Ablations (in the order of importance to contribution)**
1. L242: The authors has chosen PG19 to compute BI and model importance. Can an ablation be performed to understand the effect of calibration datasets?
   - I would like to see if model X picks layer A, B, C when using dataset 1 and U, V, W when using dataset 2. i.e whether the model has different order for BI when using different datasets as this has different implications depending on the outcome!.
2. L429: Can the authors show results on number of layers removed (or % layers removed) with respect to model size and/or task-wise for a particular performance threshold (in Appendix maybe).
   - I believe the current experiments has these insights, it's about presenting them and getting new insights when using BI i.e eg: 7B models can remove 5 layers but 13B models can remove 8 layers across tasks and/or task A, B, C can have q% layers removed but task X, Y, Z has only p% layers removed.
3. Given this falls under structured pruning, I believe more results regarding the speed-up/inference can be highlighted/commented in addition to the ones presented in Table 4.
4. Continuing from point 1 in clarifications, an ablation experiment will be helpful if we treat attention and FFN layers separately i.e remove attention layer 1, 4, 5 and FFN 2, 4, 6 etc;. Basically computing the BI after the attention block and FFN block separately; this is similar to design choice in [1], so the study can be complementary to understand the metric dependency on different blocks/layers.

**Relevant missing citations**
1. L131/Sec 2.2: The authors might have to consider citing some previous works [1, 2] which has discussed in detail and came up with similar observations regarding the final layer importance and what happens when we compress it.
2. L250: The authors can cite [1, 3, 4] which has come up with similar observations i.e compressing >30% drops the model performance.

**Citations**
1. The Cost of Compression: Investigating the Impact of Compression on Parametric Knowledge in Language Models - https://arxiv.org/abs/2312.00960
2. Fast model editing at scale - https://arxiv.org/abs/2110.11309
3. Are sixteen heads really better than one? - https://arxiv.org/abs/1905.10650
4. Compressing bert: Studying the effects of weight pruning on transfer learning - https://arxiv.org/abs/2002.08307

**Questions:**

1. The figure 2, is the experiment done to understand the effect of pre-norm vs post-norm setting keeping all the model training variables the same. If so, referring to table 11 and table 12, all the settings are not equal i.e the training tokens etc; why is that the case? An ideal comparison would be to have 2 architectures -> pre-norm vs post-norm and have all the rest the same and train it for model convergence (maybe the epochs will vary). Once these models has been trained for same loss, you can pick the hidden values and plot the similarities to observe the effect of pre vs post norm.
2. The presentation in Table 5 can be made much better in terms of highlighting which one performs better and the authors can do a better job in explaining it in Sec 4.5

**Formatting**
1. I believe Figure 1 labels are not correct and missing some information.
2. Table 3 needs better highlighting/formatting of the results; sec 4.4 can bring up important insights more clearly.
3. Table 2 can be better formatted to support the statement in L256-258 (point 2 in clarifications)

---

> ### Author Response · Authors · 2024-11-25
>
> **To clarifications**  Thank you for your suggestion. It does mean replacing the removed layers with identity matrices.
> This issue still needs to be explored. Currently, we suspect it may be related to the model. You can refer to "Scaling Law for Post training after Model Pruning", which states "From Figure 6c, We observe that the loss curve for Llama3.1-8B after width pruning is unusually flat and remains higher than that of depth pruning at the similar pruning ratio. "However, for other models, there are different conclusions.
>
> **To Additional Experiments**
> I think calibrating datasets is a very interesting thing. We used mmlu as the calibration dataset, but the results did not significantly improve the performance of mmlu. It feels that coarse-grained pruning methods like layer removal are relatively less affected by the calibration dataset.
> | Model       | PG19 Layers Removed   | MMLU Layers Removed   | MMLU (Calib PG19) | MMLU (Calib MMLU) | MMLU (Full Layers) |
> |-------------|-------------|-------------|-------------|-------------|--------------|
> | LLaMA-2-7B  | ['25', '27', '26', '24', '28', '29', '23', '22', '21'] | ['27', '25', '26', '28', '24', '23', '29', '22', '21'] | 44.5              | 44.5              | 45.9               |
> | LLaMA-2-13B | ['33', '31', '32', '30', '34', '29', '35', '28', '27', '36'] | ['31', '33', '32', '29', '30', '35', '34', '27', '28', '36'] | 48.5              | 48.5              | 55.7               |
>
> **Relevant missing citations**
> Thank you for your suggestion. Our future versions (although they may need to be resubmitted somewhere) will reference and discuss it.
>
> **To question 1**
> The settings in Table 11 correspond to Figure 2, and the settings in Table 12 correspond to Table 6. They are completely different experiments.
>
> The experiments of post-norm and pre-norm in Figure 2  are strictly controlled, with only the position of norm being different. The total number of tokens trained is different because the loss no longer converges halfway through the post-norm training.
>
> Finally, thank you again for your valuable suggestion. We will make revisions in the future. We welcome you to continue the discussion.

---

> > ### Comment · Reviewer_nyGp · 2024-11-25
> > **Rebuttal feedback**
> >
> > Thanks for the rebuttal. Most of the questions were addressed here and I would like to maintain my score.
> >
> > As the authors as pointed out, adding these experiments (and additional experiments such as bT5h Cosine experiment) as part of some ablation studies or at Appendix in addition to explanations/discussions following from previous missed citations that has come up with similar findings will definitely strengthen this paper.
> >
> > Finally, all the best with your submission.

---

### Official Review · Reviewer_Go5B · 2024-11-02

**Soundness:** 1
**Presentation:** 2
**Contribution:** 1
**Rating:** 3
**Confidence:** 5

**Summary:**

This paper introduces ShortGPT, a pruning technique for large language models (LLMs) that identifies and removes redundant layers without significantly impacting performance. The authors introduce a pruning metric called Block Influence (BI), which measures the importance of each layer by analyzing the similarity between its input and output representations. Layers with lower BI scores are considered redundant and removed. Through empirical evaluations, ShortGPT demonstrates superior performance compared to existing pruning methods, achieving high parameter reduction (around 25%) while maintaining approximately 90% of the model’s performance.

**Strengths:**

1.The proposed method of layer removal based on BI scores is straightforward, efficient, and performs comparably or better than more complex pruning methods, making it accessible and adaptable for practical use.

2.The proposed method is shown to work effectively on non-transformer models (RWKV and Mamba), indicating that the concept of layer redundancy may be applicable across various architectures, enhancing the method’s potential for broader use.

**Weaknesses:**

1.The novelty of this paper is limited, with a marginal contribution. The methodology section primarily introduces the BI pruning metric, measuring average cosine similarity between consecutive layer activations. However, [1] has already explored this, pointing out that for both Baichuan2-7B and Llama2-7B, similarity between layers 3 to 28 is typically near 1.

2.The results reported in the paper are unreliable:

a)	The tables in this paper contain numerous inconsistencies that undermine the credibility of the results. For instance, in Table 8, the SliceGPT results for Llama-2-7B and Llama-2-13B under a 20% pruning ratio are identical across PIQA, Winogrande, Arc-e, and Arc-c, yet differ for Hellaswag. Despite this discrepancy, the reported average remains inexplicably unchanged.
b)	According to Table 2, the results of the baseline methods are taken directly from [1]. Did the authors ensure a fair comparison by following the exact same settings (e.g., implementation environment, calibration dataset, benchmark script versions) as used in [1]?

c)	While the authors compared the results of ShortGPT against the baseline methods in the baseline method’s settings, they only claimed that “the same benchmarks, models and pruning ratio” are used. However, there are other settings which may affect the results, such as calibration dataset.

d)	In Appendix C, while the authors compared ShortGPT results with baseline methods SliceGPT and LLMPruner using the baseline settings, they only specified 'the same benchmarks, models, and pruning ratio' and directly referenced results from the original papers. However, additional settings, such as the calibration dataset, may also affect the outcomes. The authors should ensure all settings are consistent to enable a truly fair comparison.

e)	In Appendix C, while the authors compared ShortGPT results with baseline methods SliceGPT and LLMPruner using the baseline settings, they only specified 'the same benchmarks, models, and pruning ratio' and directly referenced results from the original papers. However, additional settings, such as the calibration dataset, may also affect the outcomes. The authors should ensure all settings are consistent to enable a truly fair comparison.

[1] Yifei Yang, Zouying Cao, and Hai Zhao. Laco: Large language model pruning via layer collapse, 2024.

**Questions:**

Apart from questions in weaknesses, there are following additional questions:

1.There are a few typos, though they do not impact the overall evaluation. For example, the incorrect y-axis label in Figure 1(b); and the denominator in the second term of Equation 4.

2.There are also some inconsistent representations, such as “LLMPru,” “LLMprun.,” “LLM Pruner,” and “LLM-Pruner.” We assume these refer to the same work, but the variations are confusing.

3.The y-axis scale in Figure 1(a) may be misleading due to its exponential nature, which could mask a significant gap in PPL between the two approaching lines.

4.Could you clarify the process used to sample instances for calculating PPL (e.g., results in Table 1)? Clarifying the sampling criteria or methodology would improve reproducibility and understanding.

---

> ### Author Response · Authors · 2024-11-25
>
> **To weakness 1**. Obviously, our starting point is different from Laco's. They believe that the removed layers also have some purpose, and in eq1, they chose to preserve the difference between the removed and retained layers and add them up. And we believe that the redundancy of these layers is very high and can be directly deleted. And the experiment also proved that direct deletion has a better effect.
>
> **To weakness 2**.  I'm sorry for making you feel like 'The results reported in the paper are unreliable'.
> a) Sorry, there were some errors in filling out and proofreading here. As you mentioned, the Hellaswag numbers in Table 8 are inconsistent with the original paper, but other indicators and averages are consistent with the original text. We will revise this section.
>
> b,c,d) I think what you said about the calibration dataset is very reasonable. Calibration datasets can indeed have an impact on the results, especially for traditional pruning methods. But please consider whether choosing a calibration dataset is more advantageous for coarser grained pruning methods or finer grained pruning methods? At such a coarse granularity for layer removal, the impact of different calibration datasets will be reduced unless the calibration dataset itself is too limited. However, this question is indeed very interesting, as mentioned by Reviewer 5T5h. It should be quite interesting to calibrate and test the performance of mmlu. And we list below:
> | Model       | PG19 Layers Removed   | MMLU Layers Removed   | MMLU (Calib PG19) | MMLU (Calib MMLU) | MMLU (Full Layers) |
> |-------------|-------------|-------------|-------------|-------------|--------------|
> | LLaMA-2-7B  | ['25', '27', '26', '24', '28', '29', '23', '22', '21'] | ['27', '25', '26', '28', '24', '23', '29', '22', '21'] | 44.5              | 44.5              | 45.9               |
> | LLaMA-2-13B | ['33', '31', '32', '30', '34', '29', '35', '28', '27', '36'] | ['31', '33', '32', '29', '30', '35', '34', '27', '28', '36'] | 48.5              | 48.5              | 55.7               |
>
> **To question 1&2**  Thanks for pointing it, we will revise.
>
> **To question 3**
> The y-axis of Table 1 (a) you mentioned may be misunderstood, but we believe this does not create a misunderstanding. We just want to prove that except for a few layers, the influence of most layers is very small. It is not a critical issue to determine which layer has a slightly larger or smaller impact, 15 or 16.
> 5. The sampling method for calculating PPL is to randomly select 10 text segments of 1k length from each piece of data in PG19, and calculate PPL. The final PPL is the average of the PPLs of all these segments.
>
> **To question 4**
> The sampling method for calculating PPL is to randomly select 10 text segments of 1k length from each piece of data in PG19, and calculate PPL. The final PPL is the average of the PPLs of all these segments.

---

### Official Review · Reviewer_oU5S · 2024-11-03

**Soundness:** 2
**Presentation:** 2
**Contribution:** 2
**Rating:** 3
**Confidence:** 4

**Summary:**

The paper presents a layer pruning method specifically designed for LLMs. It begins by assessing layer importance using the proposed Block Influence metric, which quantifies importance based on the variation in features between consecutive layers. Subsequently, the method iteratively removes the least significant layers until the target sparsity level is achieved.

**Strengths:**

The proposed method is straightforward to implement and requires only forward passes, making it computationally efficient. This efficiency allows it to be applied to larger models within a limited computation budget, unlike competing methods that require the additional overhead of computing gradients or Hessians.

**Weaknesses:**

1. The method hinges on the skip-connection nature of LLM layers, specifically, $x_{L+1} = x_L + f(x_L)$. Consequently, this reliance could limit its applicability compared to competitors that use gradients and are generally not constrained by model architecture.

2. The analysis on layer redundancy in Appendix A is unconvincing. Specifically, it misinterprets Lemma 1, which states that $||x_L|| = O(\sqrt{L})$ rather than $||x_L|| = \Theta(\sqrt{L})$. Furthermore, the authors assert that $f_L(x_L) = O(1)$ without solid mathematical proof. In the Attention example, the authors omit $W_o$ initially but include it later without explanation. Additionally, in (4), it is unclear why the $\ell_2$ norm of $||x_{L+1}||_2$ diminishes. Overall, the analysis in this section lacks clarity and cohesion.

3. The paper does not compare its method with recent gradient-based pruning techniques such as SLEB [1] and Shortened Llama [2], and it lacks a discussion on the related work by Gromov et al. [3].

4. In Table 2, for WSC, why are the accuracies of all Dense models the same at $50\%$? Additionally, the accuracies for ShortGPT-pruned Llama2-13B and Baichuan2-7B are also all $50\%$ on WSC.

5. Section 4.5 lacks a comparison with other pruning methods. It remains unclear if these other methods, when applied to quantized models, could potentially outperform ShortGPT.

6. Section 4.6 also lacks a comparison with other pruning methods.

**Overall Assessment**

In summary, I don't think the technical contribution of this paper is sufficient for ICLR.

[1] Song, Jiwon, et al. "SLEB: Streamlining LLMs through Redundancy Verification and Elimination of Transformer Blocks." arXiv preprint arXiv:2402.09025 (2024).

[2] Kim, Bo-Kyeong, et al. "Shortened llama: A simple depth pruning for large language models." arXiv preprint arXiv:2402.02834 11 (2024).

[3] Gromov, Andrey, et al. "The unreasonable ineffectiveness of the deeper layers." arXiv preprint arXiv:2403.17887 (2024).

**Questions:**

Please see Weaknesses and address them.

---

> ### Author Response · Authors · 2024-11-25
> **Response to Reviewer oU5S**
>
> **To weakness 1** At present, most large language models are based on skip connections, and even most are based on pre-norm. Whether it is a LLM with a transformer architecture or a non transformer architecture , it is a very general setting. As an aside, can we train a large language model without skip connections or their variants?
>
> **To weakness 2**  (1) Lemma 1 also gives upper and lower bounds, so it is Θ (L) instead of O (L). (2) Sorry, there is some typographical error here. The  q in front of it is actually the  o, and the ℓ 2 norm of $| | x_{L+1} | |_2$ was accidentally ignored due to a typographical error. After correcting the typo, do you think there are any issues with the proof?
>
> **To weakness 3** .The jobs you mentioned are working at the same time as ours, some one or two weeks earlier and some one or two weeks later. The essence of the content is not much different. Undoubtedly, these works all point to the significant layer redundancy in existing large language models.
>
> **To weakness 4** We have checked our review code and it is indeed like this. All tests were conducted using the OpenCompass tool. Even if the WSC dataset is excluded, it does not affect our conclusion.
>
> **To weakness 5&6** We did not compare other methods in 4.5 and 4.6 for these reasons. Firstly, it is just an analysis section. We just want to demonstrate the layer redundancy that exists in large language models. We just want to demonstrate the potential of layer removal methods in LLM. I think we have basically achieved our goal.

---

> > ### Comment · Reviewer_oU5S · 2024-11-26
> >
> > Thank you for your clarification.
> >
> > W1: My main point is that other gradient-based methods do not assume specific model architectures. How can you be confident that LLMs will consistently adhere to the current mainstream architecture?
> >
> > W2: Thank you for clarifying the typo. I hope the manuscript can be thoroughly polished before submission. Regarding my additional questions, please refer to my initial comments and ensure all queries are adequately addressed.
> >
> > W3: The mentioned works have been accepted at reputable conferences such as ICML 2024. At a minimum, you should cite and discuss these studies. Overlooking published work is not a recommended practice. Regarding the statement, "The essence of the content is not much different," what is your **unique contribution** then?
> >
> > I will maintain my score.

---

### Meta-Review · Area_Chair_w9eH · 2024-12-16

**Metareview:**

This paper studies a layer pruning method using a layer-importance score called Block Influence (BI) based on the variation in representations between input and output layers. Empirical evaluations demonstrate the effectiveness of the method.

Main weaknesses mentioned by reviewers:
- Reliance on specific model architectures
- Limited comparison to existing work; limited novelty
- Unreliable results due to inconsistent comparisons

**Additional Comments On Reviewer Discussion:**

During the rebuttal, the authors clarified several places with typographical errors in theories and inconsistent numbers in experiments. I believe the paper will benefit from another round of review to improve.

---

### Decision · Program_Chairs · 2025-01-22

Reject